# Non-Indigenous Canadians' and Americans' moral expectations of Indigenous peoples in light of the negative impacts of the Indian Residential Schools

**Mackenzie J. Doiron[1], Nyla Branscombe[2], Kimberly Matheson**[3,4] *

**1** Department of Psychology, Carleton University, Ottawa, Ontario, Canada, **2** Department of Psychology, University of Kansas, Lawrence, Kansas, United States of America, **3** Department of Neuroscience, Carleton University, Ottawa, Ontario, Canada, **4** Institute of Mental Health Research, University of Ottawa, Ottawa, Ontario, Canada

* KimMatheson@cunet.carleton.ca

## Abstract

The historical trauma associated with the Indian Residential School (IRS) system was recently brought to the awareness of the Canadian public. Two studies investigated how the salience of this collective victimization impacted non-Indigenous Canadians' expectations that Indigenous peoples ought to derive psychological benefits (e.g., learned to appreciate life) and be morally obligated to help others. Study 1 found that modern racism was related to perceptions that Indigenous peoples psychologically benefitted from the IRS experience, which in turn, predicted greater expectations of moral obligation. Study 2 replicated the relations among racism, benefit finding, and moral obligation among non-Indigenous Canadians (historical perpetrators of the harm done) and Americans (third-party observers). Americans were uniquely responsive to a portrayal of Indigenous peoples in Canada as strong versus vulnerable. Factors that distance observers from the victim (such as racism or third-party status) appear to influence perceptions of finding benefit in victimization experiences and expectations of moral obligation.

## Introduction

Canada's Truth and Reconciliation Commission (TRC) released its final report in 2015 documenting the historical and persisting ramifications of the Indian Residential Schools (IRSs) for Indigenous peoples and communities [1]. The report brought to the forefront the harm perpetrated against Indigenous peoples by the Canadian government and the churches that operated the schools, while at the same time, giving voice to their stories, legitimizing their claims of injustice, and advocating for a pathway forward. For these claims to mobilize government agencies to meet the calls to action put forward in the TRC report, there needs to be acceptance from the Canadian public. Yet, when past harms perpetrated against a group are highlighted, it has been found that victim group members are often perceived to have psychologically

https://doi.org/10.5683/SP2/XIPNFF, https://doi.
org/10.5683/SP2/4Z8FDK.

**Funding:** KM, NB Grant number 435-2018-1220
Social Sciences and Humanities Research Council
of Canada https://www.sshrc-crsh.gc.ca The
funders had no role in study design, data collection
and analysis, decision to publish, or preparation of
the manuscript.

**Competing interests:** The authors have declared
that no competing interests exist.

benefitted from the experience (e.g., learned to appreciate life more) [2]. Because they know what it means to experience undeserved suffering, members of victimized groups are often expected to be better, stronger people, and are held to a particularly high standard of moral conduct (i.e., have an obligation to help or not harm non-adversarial others) [3–5].

In revealing the atrocities against Indigenous children that occurred in IRSs, the TRC report triggered a process of reflection and wider meaning-making for non-Indigenous Canadians. Thus, an unintended consequence of the heightened salience of the legacy of the IRSs may be to instigate a psychological process among historical perpetrator group members (i.e., non-Indigenous Canadians) wherein they attribute psychological benefits to the victims, and subsequently hold them to a higher moral standard of behavior. To assess this possibility, two studies were conducted to explore this relationship. We were particularly interested in whether modern racism further predicted the expression of such expectations, and whether they would predominate when non-Indigenous Canadians were encouraged to reflect on the meaning of the IRSs for the lives of Indigenous peoples today (Study 1). As the Canadian media conveying the impacts of the IRSs varied in terms of its emphasis on the continued inequities experienced by Indigenous peoples and/or their perseverance and resistance to colonialist actions, a second study was conducted to assess the effects of highlighting the strengths versus continued vulnerability of Indigenous peoples on non-Indigenous Canadians' perceptions of the implications of the IRS system for benefits derived and moral obligations of Indigenous peoples (Study 2).

## The Indian Residential Schools in Canada

Beginning in the late 1800s, the Canadian government officially established the IRSs, in partnership with the churches, as an explicit strategy to assimilate Indigenous peoples into the dominant Euro-Christian culture. Children were often forcibly removed from their home communities to attend IRS. Of the over 150,000 children who attended, a known 4,090 would never return home alive (although it is believed that the number is significantly higher), and often family were never informed of what happened to their child [1]. The IRS students faced multiple forms of maltreatment, including neglect, as well as physical, psychological, and sexual abuse. While at the IRSs, cultural practices and use of Indigenous languages were punitively discouraged by school staff. As a result of these experiences, IRS survivors often returned home traumatized by abuse, having lost their cultural identity and their language, rendering them outsiders in their own communities, and leaving them unequipped and unwelcome to go elsewhere. It was only in 1996 that the doors of the last remaining IRS were closed.

The nearly 150-year effort to eradicate Indigenous culture, while denying full acceptance of Indigenous peoples within Canadian society, resulted in numerous intergenerational issues that continue to create negative life conditions for IRS survivors, their descendants, and their communities [1]. These life conditions include socioeconomic issues (poverty, unemployment, lower education), greater exposure to poor living conditions (lack of housing, food insecurity) and to severe life stressors (criminal victimization, incarceration, foster care), as well as greater mental (suicide, trauma, depression, learning difficulties, substance use) and physical (diabetes, tuberculosis) health challenges [6, 7]. It has been suggested that Indigenous peoples are the most systemically disadvantaged group in Canada, and instances of continued racism are 'alarmingly high' [8].

With the extensive media coverage following the release of the TRC report, many Canadians became aware of the IRSs and their legacy for the first time. As with Truth and Reconciliation efforts globally, unmasking the past harms perpetrated against Indigenous peoples makes it possible to hold a public discourse and dialogue that could form the foundation of healing, reconciliation, and social equality [9]. However, the salience of the victimization of Indigenous

peoples in Canada might also bring to the fore motivational biases that encourage non-Indigenous Canadians to justify past actions, or at the very least, to derive ingroup-protective meaning going forward [10].

## Expectations of Indigenous benefit finding and moral obligations

Predominant among the factors that influence responses to victims is a strong motivation to maintain a belief in a just world [11]. This motivation has been found to result in victim derogation, especially when the victimization experience was highly impactful and traumatic [10, 11]. In order to restore just world beliefs, observers may construe victim suffering as compensated by moral virtue, and generate perceptions that the victims or their descendants, have acquired 'benefits' as a result of their experience [2, 5]. In effect, observers are motivated to find meaning in the experience of members of a victimized group by seeing them as having grown and thereby benefited from their experience. Moreover, because of their past suffering (e.g., the Jewish Holocaust), the victim group should understand what it means to suffer and they (or their descendants) can be expected to be better, stronger people. As a result of gaining such understanding, victims are held to higher standards of moral conduct [3, 4]. For example, when the prior suffering of individuals who had experienced victimization in the past (e.g., adult victims of child abuse) was salient, observers were more likely to view them as morally obligated to help others, and to not cause harm [5]. This expectation was especially likely to emerge when observers sought meaning in the experience for the victim, rather than for the perpetrator. Likewise, members of victimized minority groups were expected to be more tolerant of other persecuted minorities when their own history of discrimination was salient [4], and to be more committed to social justice [12].

Although past research regarding victim moral obligation has focused on the perceptions of third-party observers (i.e., members of groups that were not directly involved, historically or currently, in the perpetration of harm), there is reason to believe that if the past harm done (or continued inequities) call into question the moral status of the group that perpetrated the harm, this can represent a powerful threat that elicits a defensive response [13, 14]. A need to re-assert moral superiority among members of the perpetrator group may be especially likely if the harm experienced by the victim is severe or traumatic and their suffering is ongoing [15]. To restore their sense of moral rightness and affirm their own collective identity, members of the perpetrator group may justify their group's actions or minimize the suffering of the victims [16]. In this regard, although there exist strong social norms that discourage direct victim blame, 'modern racism' [17] is often expressed by attributing differential values and attitudes that serve to justify group inequities, or attitudes that reflect that enough change has occurred to reasonably alleviate inequities. In order to restore their own belief that the world is just, when racial injustices are made salient, modern racist biases might elicit indifference to the victim group's suffering [18], and bolster beliefs that they have benefited from their past victimization. Indeed, a good example of this reaction was evidenced by the response of the now notorious Canadian Senator, Lynn Beyak who, when confronted with the abuses perpetrated against Indigenous children in the IRSs, argued that 'there are shining examples from sea to sea of [Indigenous] people who owe their lives to the schools' and chastised the TRC for 'not focusing on the good' of 'well-intentioned' IRS staff [19].

## The present investigation

In the present investigation we sought to determine whether an unintended consequence of the heightened salience of the past victimization of Indigenous peoples (and in particular, the legacy of the Indian Residential Schools) was to instigate a process among historical

perpetrator group members (non-Indigenous Canadians) wherein they would attribute psychological benefits to the victims, and subsequently hold them to a higher moral standard of behavior. Past research documenting this pattern of perceptions has been conducted from the perspective of third-party observers. It was possible that historical perpetrator group members might either be more sympathetic (especially in light of the continued discrimination encountered by Indigenous peoples in Canada), or conversely, more defensive, particularly among those who hold racist beliefs. Study 1 assessed whether non-Indigenous Canadians' reflection on the implications and lessons learned from the IRS system (search for meaning) resulted in heightened benefit finding and perceived moral obligations of Indigenous peoples. The possibility that these perceptions would be especially evident among those who expressed high levels of modern racism was tested. Study 2 assessed whether the responses of non-Indigenous Canadians differed from those of Americans, who ostensibly represented a third-party observer group, and whether explicitly highlighting the current strengths and perseverance of Indigenous peoples increased the extent to which they were viewed as morally obligated to others.

## Study 1

It has consistently been demonstrated that observers are more likely to perceive victims to have benefitted from their experience, and as a result hold them to a higher standard of moral behaviour. Such perceptions were particularly likely to occur after observers tried to make sense of the implications and lessons learned by victims from their past (or historical) experience [3, 4]. Presumably, the salience of the victim's suffering triggered observers' need to reassert just world beliefs by viewing the victim as finding benefit in their past experience. Ironically, but consistent with just world beliefs, these greater perceived benefits and moral obligations were less evident when observers focused on the meaning of the past transgression for members of the perpetrator group (or compared to a control group that did not reflect on the meaning of the experience). However, unlike third-party observers, it is possible that when perpetrator group members seek meaning *for themselves* in the past harm done by their group (i.e., consider what the events imply about their own group's actions and moral standing), they might attribute greater benefits and obligations to the victim group as a defensive strategy for protecting their own group identity [14] and reducing the need for retributive justice [20].

Study 1 assessed non-Indigenous Canadians' perceptions of Indigenous peoples in light of the salience of the IRS experience. It was hypothesized that when presented with information regarding the IRS system and asked to derive meaning from this past suffering for Indigenous peoples (vs. meaning for Canadians vs. a control group that was not asked to reflect on the meaning of the IRS experience), non-Indigenous Canadians would perceive Indigenous peoples as deriving greater benefit and having greater moral obligation (Hypothesis 1). We also expected that greater perceived benefit finding would mediate the relation between the meaning-making condition and perceived Indigenous moral obligation (Hypothesis 2). Finally, it was hypothesized that modern racism would exacerbate the extent to which non-Indigenous Canadians perceived greater benefit finding among Indigenous peoples and, in turn, this would predict greater moral obligation (Hypothesis 3).

### Materials and methods

**Participants and procedure.**   Students from a Canadian university ($N$ = 102; 66 females, 36 males; $M_{age}$ = 20.22, $SD$ = 5.32 years) were recruited to participate in a study on their opinions regarding the experiences of Indigenous peoples in Canada, particularly in relation to the IRS system. Based on effect sizes associated with the effects of meaning-making focus on

perceived victim benefit-finding and moral obligation reported in Branscombe et al. [3] ($\eta^2$s ranged from .08 to .11), this sample size provided sufficient power to detect differences at $\alpha$ = .05 and $\beta$ = .84. Criteria for inclusion were that participants be Canadian citizens and not self-identify as Indigenous.

After providing written informed consent, participants were randomly assigned to one of three conditions using the randomizer feature in Qualtrics, an online survey platform. All participants read a brief passage about the abuses incurred by the IRS system. The passage provided them with the historical facts regarding the IRSs (e.g., "130 residential schools operated through much of Canada from the mid-1800s until the last school closed in the 1990s. By the 1930s, approximately 75% of First Nations children attended these schools, as did many Métis and Inuit children."). In addition, the victimization of many of the students who attended the schools was described (e.g., ". . .these children were often subjected to neglect, emotional, physical, and sexual abuse").

After reading the passage, participants were given 10 minutes to write about "the implications or lessons derived from the Indian Residential Schools". To manipulate the focus of the meaning of the IRS experience, participants either wrote about the meaning derived for Indigenous peoples, for Canadians, or they were not asked to reflect on the meaning of the IRSs (control). Participants then completed the outcome measures and a modern racism scale, were debriefed, and provided with investigator contact information. Both studies in this investigation were approved by the Carleton University Research Ethics Board-B (#10–5303).

**Measures.**   All of the measures assessed participants' agreement with item statements on a 7-point scale ranging from 1 'strongly disagree' to 7 'strongly agree'. In all instances, average scores were calculated, with relevant items reverse scored.

Outcome measures were adapted from Branscombe et al. [3]. *Perceived benefit finding* for victims of the IRS system and their descendants entailed a five-item measure (e.g., "Because of their victimization history, Indigenous peoples should appreciate their lives more") (Cronbach's $\alpha$ = .86; $M$ = 4.35, $SD$ = 1.26). *Perceived moral obligations* for Indigenous peoples in light of their historical victimization (e.g., "A central lesson from the Indian Residential School experience is that Indigenous peoples must take care not to inflict suffering upon other people") was assessed using five items (Cronbach's $\alpha$ = .80; $M$ = 5.06, $SD$ = 1.10).

Adapted from McConahay [17], 15 items assessed self-reported *modern racism* (e.g., "Indigenous peoples should stop complaining about the way they are treated, and simply get on with their lives") ($\alpha$ = .92; $M$ = 3.00, $SD$ = 1.07). Levels of racism reported did not significantly vary across meaning-making conditions.

To evaluate level of knowledge about the IRS system and its legacy, participants were asked three questions developed for this study including "How aware are you of the following? (1) the Indian Residential School system; (2) Canada's statement of apology; and (3) the Truth and Reconciliation Commission of Canada." Responses were made using 3-point rating scales ranging from 1 'not at all aware' to 3 'very aware' ($\alpha$ = .75). The mean score (averaged across these items) was at the midpoint ($M$ = 2.06, $SD$ = 0.54), suggesting that, on average, participants were moderately aware of the issues.

## Results

To determine the effects of the meaning-making condition on perceptions of benefit finding and Indigenous peoples' moral obligations, and whether these perceptions were moderated by self-reported modern racism, hierarchical regressions were conducted. For each outcome variable (benefit finding, moral obligation), meaning-making condition was entered on the first step (as a multi-categorical variable, Helmert coding was employed wherein X1 assessed

differences between focusing on Indigenous peoples vs. Canadians and the control condition (combined), and X2 compared responses when participants derived meaning for Canadians vs. the control condition), followed by modern racism, and their interaction (cross-products) on the last step. Neither of the contrasts reflecting the main effect of meaning-making condition or the interaction terms with modern racism were significant, $Fs<1$ (means in Table 1). However, irrespective of condition, modern racism was a significant predictor of greater perceived benefit finding, $b = 0.50$, $SE = 0.11$, $p < .001$, and Indigenous moral obligation, $b = 0.21$, $SE = 0.10$, $p = .043$.

To assess whether benefit finding mediated the relationship between modern racism and perceived moral obligation of Indigenous peoples, the PROCESS macro (Version 3.3), applying model 4 [21] was used. The macro was set to use bootstrapping procedures with 5000 resamples. As seen in Fig 1, the total effect of modern racism in relation to Indigenous moral obligation was significant, $c = 0.23$, $SE = 0.10$, $CI_{.95}$ [.03, .43]. However, with the inclusion of benefit finding as a mediator, the direct effect was not significant (i.e., 95% CI contained 0), $c'$ = 0.07, $SE = 0.11$, $CI_{.95}$ [-.14, .28]. As benefit finding was related to expectations of moral obligation, $b = 0.31$, $SE = 0.09$, $CI_{.95}$ [.14, .49], the indirect mediated effect of racism on Indigenous moral obligation through benefit finding was significant, $ab = 0.15$, $SE = 0.05$, $CI_{.95}$ [.05, .27], suggesting that benefit finding can account for the relationship between modern racism and Indigenous moral obligations. Given that these data were correlational, all other possible orders of this mediation model were tested, but none of the alternative directional pathways was significant.

## Discussion

As hypothesized, after making salient the victimization of Indigenous peoples in the IRS system, non-Indigenous Canadians who perceived Indigenous peoples as deriving some benefit from the experience (e.g., learning to be 'stronger as a people') were more likely to regard Indigenous peoples as obligated to be concerned about the well-being of others (e.g., 'to ensure that they never act toward others in the same way'). Moreover, such perceptions were more likely to be endorsed by those whose attitudes toward Indigenous peoples could be construed as racist, and it was as a result of being able to frame the IRS experience as beneficial to Indigenous peoples that racism contributed to holding Indigenous peoples to a higher standard of behaviour (mediation model).

Study 1 did not fully replicate prior research [3, 5] that led us to expect that perceived benefits and moral obligations would be greater when participants concentrated on finding meaning in the IRS experience for Indigenous peoples (victim group), relative to searching for meaning for Canadians (the perpetrator group), or in the control group. This was not simply an issue of statistical power, as effect sizes in the present research were very small ($\eta^2$s were .009 and .012, respectively), whereas they were more moderate in previous research (e.g., $\eta^2$s

**Table 1. Means (standard deviations) for outcome variables as a function of meaning-making condition in Study 1.**

|  | Meaning-Making Condition | | |
|---|---|---|---|
|  | **Indigenous Peoples** | **Canadians** | **Control** |
| Benefit Finding | 4.21 (1.11) | 4.33 (1.60) | 4.50 (1.06) |
| Moral Obligation | 4.86 (1.21) | 5.09 (1.14) | 5.22 (0.98) |

Responses could range from 1 to 7.

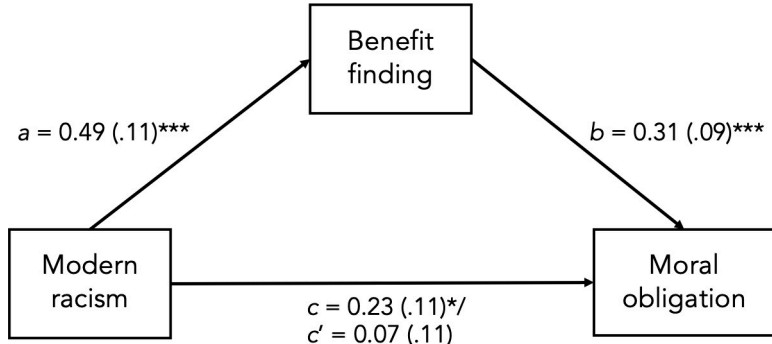

**Fig 1. Unstandardized path coefficients (*SE*) for the mediated relationship between modern racism and attributions of moral obligation to Indigenous peoples through perceptions of benefit finding from the IRS experience in Study 1.** * $p < .05$; *** $p < .001$.

were .10 and .09, respectively in Branscombe et al. [3]. In contrast to previous research, participants in the present study were members of the group that represented the historical perpetrator of the IRS system and the harm done to Indigenous peoples. Thus, it is possible that the salience of the victimization of Indigenous peoples called into question the moral status of participants' own collective identity [14, 15]. For this reason, modern racist attitudes toward Indigenous peoples may have played a prominent role by motivating some participants to justify the IRSs through their beliefs that benefits were bestowed by the IRS system. Alternatively, it is possible that the continued inequities affecting Indigenous peoples were sufficiently salient to non-Indigenous Canadians that they readily understood that the IRS experience could not be construed as beneficial. Indeed, when victims are viewed as still suffering, observers are less likely to hold them to a higher moral standard [4].

## Study 2

Following the release of the TRC report, the Canadian media portrayal of Indigenous peoples oscillated between depicting them as strong, resilient, and having persevered in spite of their victimization history, and framing them as vulnerable and continuing to suffer from collective and historical trauma. A strength-based framework is reminiscent of the benefits observers are expected to attribute to victims as they reflect on the meaning and implications of victims' experience (i.e., 'what doesn't kill you makes you stronger'). If so, while this understanding might enhance intergroup respect [22], it may also lead to a diminished perception of the severity of the struggles that continue to be faced by the victimized group, a belief that equality has been achieved and further entitlements are not warranted, and an expectation of greater moral obligation on the part of victim group members. In contrast, when the media presents a vulnerability-based framework, this may evoke perceptions that victimized group members are inherently weak and continue to suffer beyond the event. This discourse may serve to engage sympathies toward the victimized outgroup, but it also risks overemphasizing the need for assistance. This may cause observers to endorse a more paternalistic relationship and diminish perceptions that victimized outgroup members have learned from their experience and are 'ready' to be treated as equals [23, 24]. Study 2 aimed to examine whether explicitly framing Indigenous peoples as strong despite their victimization experience, versus continuing to suffer as a result of it, influenced perceived benefit finding, and hence greater moral obligation.

To assess whether non-Indigenous Canadians' historical perpetrator status contributed to their reactions to Indigenous peoples, in Study 2, the responses of Americans were also

assessed. Although Americans' treatment of Indigenous peoples in the United States has also been rife with racism, with many similar actions to eliminate and assimilate Native Americans, this issue has not reached the awareness of many Americans, including the parallels to the Canadian tactics [25]. Owing to the profound sociocultural and geographic similarity to Canada, American participants offered a useful comparison group. Despite the similarities in the types of harms perpetrated against Indigenous peoples, by explicitly framing the injustices as being perpetrated by Canadians, Americans may be able to express their views on the treatment of Indigenous peoples while distancing themselves. Thus, much like previous research manipulating third-party observers' perceptions of the meaning of events for victims versus perpetrators [3, 5], Americans' reactions may be sensitive to the salience of the achieved strengths versus continued vulnerability of Indigenous peoples in Canada.

In Study 2, we expected that when the achieved strengths (rather than continued suffering) of Indigenous peoples were made salient, greater benefit finding and Indigenous moral obligation would be perceived; given their psychological distance (third-party observer status), this effect would be strongest among Americans (Hypothesis 1). Among both national groups, modern racism was expected to predict expectations of greater Indigenous moral obligations, and it was expected that benefit finding would mediate this relation (Hypothesis 2). Finally, the possibility that this mediated relation would be less evident when Indigenous peoples were perceived to experience continued discrimination (reducing perceptions that they derived psychological benefit) was assessed (Hypothesis 3).

## Materials and methods

**Participants and procedure.** Participants were recruited via Amazon's MTurk platform. The experiment was listed as a human information task (HIT) that MTurk workers could complete in exchange for $5 CAD. Of the 142 (all non-Indigenous) participants (69 Canadians; 73 Americans), 68 were female, and 73 were male (1 did not disclose gender). Participant ages ranged from 19 to 64 years ($M = 32.96$, $SD = 9.55$). Based on an anticipated effect size of $\eta^2 = .10$ [2], this sample size provided sufficient power ($\beta = .91$) to detect differences at $\alpha = .05$.

After providing informed consent online (by clicking accept and proceeding to the survey page), participants were randomly assigned to one of the two discourse conditions (strength $n = 71$; vulnerability $n = 71$). The discourse manipulation entailed having participants read the background information regarding the IRS experience. At the end, a conclusion was drawn regarding the current status of Indigenous peoples that either highlighted their *strengths* ("many Indigenous peoples and communities in Canada have demonstrated considerable strength and resilience. . . the fortitude to experience such trauma, but continue to survive and reclaim their core identity. They have been, and continue to be, strong advocates for the recognition of their rights.") or *vulnerabilities* as a result of the IRS experience ("many Indigenous peoples and communities in Canada continue to suffer. The cultural trauma experienced by generations has resulted in a loss of their core identity, which has caused many social and personal problems. Indigenous peoples in Canada continue to be in a process of healing."). Participants were given 10 minutes to write about the meaning of the IRS experience for Indigenous peoples (as in the victim focused meaning-making condition in Study 1). They then completed the same measures as in Study 1. In addition, to assess perceptions of continued victimization, an item to assess whether "Indigenous peoples continue to encounter discrimination as a result of the policies of the Canadian government" was included, rated along a 1 'strongly disagree' to 7 'strongly agree' scale.

## Results

**Descriptive statistics.** Levels of modern racism toward Indigenous peoples that were expressed by Americans ($M$ = 3.17, $SD$ = 1.20) were comparable to those of non-Indigenous Canadians ($M$ = 3.03, $SD$ = 1.26), $t(140)$ = 0.68, $p$ = .497. There was no difference in levels of racism as a function of discourse condition. As expected, Americans' awareness of the IRS system ($M$ = 1.47, $SD$ = 0.46) was significantly lower than that of Canadian participants ($M$ = 2.16, $SD$ = 0.63), $t(140)$ = -7.49, $p$ < .001, but as seen in Table 2, American ($M$ = 5.32, $SD$ = 1.25) and Canadian participants ($M$ = 5.28, $SD$ = 1.71) were equally likely to agree that Indigenous peoples continue to encounter discrimination, $t(140)$ = 0.16, $p$ = .874.

**Main analyses.** Hierarchical linear regression analyses were conducted wherein discourse condition (vulnerable coded 0 and strength coded 1), nationality (Americans coded 0 and Canadians coded 1), and modern racism (continuous) were entered sequentially, followed by the two-way interaction terms (cross-products), and the three-way interaction on the final step. The main effect of discourse condition was not significant for either benefit finding, $F(1,135)$ = 2.07, $p$ = .152, $\eta^2$ = .015, or moral obligation, $F$<1, $\eta^2$ = .002. However, as seen in Table 2, Americans were more likely to perceive Indigenous peoples to find benefit from the IRS experience, $F(1,135)$ = 15.42, $p$ < .001, $\eta^2$ = .102, and held them to a higher moral obligation, $F(1,135)$ = 9.78, $p$ = .002, $\eta^2$ = .068. In addition, modern racism was a significant predictor of higher levels of perceived benefit finding, $b$ = 0.28, $SE$ = 0.09, $F(1,135)$ = 12.14, $p$ = .001, and Indigenous moral obligation, $b$ = 0.26, $SE$ = 0.10, $F(1,135)$ = 9.71, $p$ = .002.

None of the two-way or three-way interactions were significant predictors of Indigenous moral obligations. The three-way interaction between discourse condition, nationality, and racism on benefit-finding was significant, $b$ = 0.80, $SE$ = 0.37; $F(1,135)$ = 5.65, $p$ = .019, $\eta^2$ = .040. Simple interaction tests indicated that, as hypothesized, Americans' perceptions varied as a function of the discourse condition, $F(1,134)$ = 4.37, $p$ = .038, whereas those of Canadians did not, $F(1,138)$ = 1.04, $p$ = .308. As seen in Fig 2, simple slope analyses assessing the moderating role of racism (1 $SD$ above and below the mean) on the effects of discourse on benefit finding indicated that at lower levels of racism, Americans perceived Indigenous peoples to have benefited from their past experience when they were depicted as strong, in comparison to when their continued vulnerability was emphasized. There was no effect of discourse condition among Americans with higher levels of modern racism, just as the perceived benefits under different discourse conditions did not vary among Canadian participants. Rather, under these conditions, benefit finding was solely predicted by greater modern racism. In short, the discourse conveying Indigenous peoples as strong versus vulnerable did not affect the perceptions of perpetrator group members. Among third-party observers, only those who did not report racist beliefs were sensitive to the discourse manipulation, perceiving strength only when such strengths were made salient. Indeed, among both non-Indigenous Canadians and Americans, higher levels of modern racism were associated with being less likely to perceive

**Table 2. Means (standard deviations) for outcome variables by nationality (Americans vs. Canadians) and discourse condition (highlighted Indigenous strengths vs. vulnerability) in Study 2.**

|  | Americans | | Canadians | |
|---|---|---|---|---|
|  | **Strength** | **Vulnerability** | **Strength** | **Vulnerability** |
| Benefit Finding | 5.40 (1.17) | 4.81 (1.44) | 4.27 (1.43) | 4.22 (1.27) |
| Moral Obligation | 4.94 (1.11) | 4.88 (1.46) | 4.27 (1.53) | 4.09 (1.59) |
| Ongoing Discrimination | 5.51 (1.10) | 5.13 (1.36) | 5.31 (1.80) | 5.24 (1.62) |

Mean scores could range from 1 to 7.

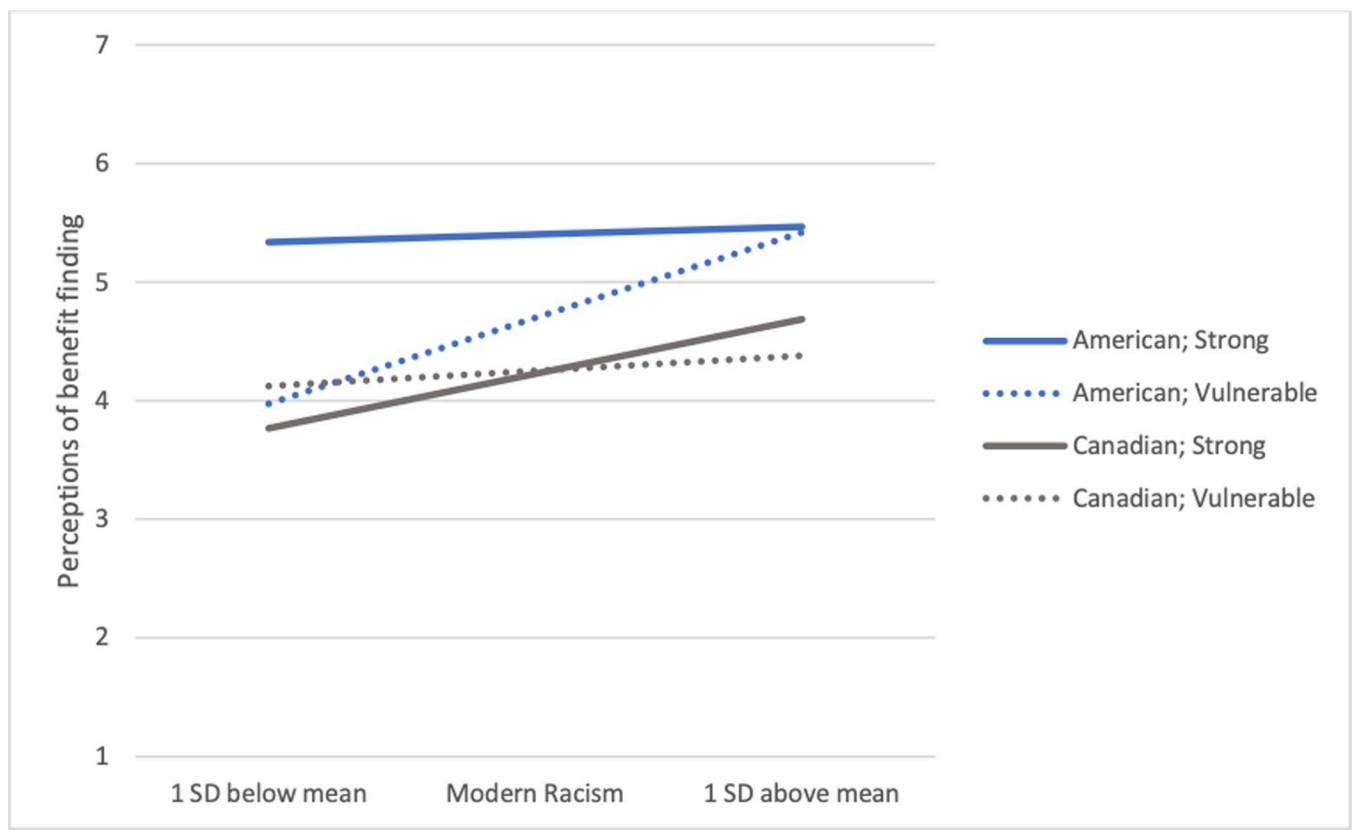

**Fig 2. Perceptions of Indigenous peoples finding benefit from the IRS experience as a function of discourse condition (strong vs. vulnerable), nationality of perceiver (American vs. Canadian) and perceivers' level of modern racism in Study 2.**

Indigenous peoples as the victims of continued discrimination, $b = -0.78$, $SE = 0.14$, $F(1,134) = 29.27$, $p < .001$.

The analysis assessing whether the relation between racism and Indigenous moral obligation was mediated by benefit finding indicated that, as in Study 1, racism was a significant predictor of benefit finding, $a = 0.34$, $SE = 0.09$, $CI_{.95}[.15, .52]$, which, in turn, was a significant predictor of Indigenous moral obligations, $b = 0.74$, $SE = 0.06$, $CI_{.95}[.61, .86]$. The total effect of racism on Indigenous moral obligations was significant, $c = 0.32$, $SE = 0.10$, $CI_{.95}[.12, .51]$. After controlling for benefit finding, the direct effect was no longer significant, $c' = 0.07$, $SE = 0.07$, $CI_{.95}[-.08, .21]$, whereas the indirect effect of racism on Indigenous moral obligations through benefit finding was significant, $ab = 0.25$, $SE = 0.07$, $CI_{.95}[.11, .39]$. Thus, as in Study 1, benefit finding mediated the relation between modern racism and Indigenous moral obligation. All other possible orders of this mediation model were tested, but none of the alternative directional pathways was significant. Nationality was not a significant moderator of the mediated model.

The possibility that the perception of continued victimization moderated the mediated relations between racism, benefit finding, and moral obligation was evaluated using the PROCESS macro model 7 [21]. The moderated mediation model was significant, Index = 0.067, $SE = 0.043$, $CI_{.95}[.001, .17]$. In effect, the mediated model was not significant when Indigenous peoples were perceived to encounter lower levels of discrimination, $ab_{-1SD} = 0.10$, $SE = 0.11$, $CI_{.95}[-.11, .32]$, but became stronger with greater perceptions of continued victimization, $ab_{+1SD} = 0.30$, $SE = 0.13$, $CI_{.95}[.09, .58]$. It appears that if the threat to just world beliefs has

been diminished by perceptions that ongoing discrimination is absent, the need to engage in benefit finding is likewise lessened.

**Content analysis of written responses to meaning-making task.** Given the differential responses of non-Indigenous Canadian and American participants to the discourse manipulation, a post hoc content analysis was conducted of participants' written responses to the meaning-making task for which they reflected on the implications and lessons learned for Indigenous peoples. On average, the written responses were 871.6 characters in length (*SD* = 429.5, ranging from 137 to 2000 characters), and there were no significant differences in the length of responses as a function of either discourse condition or nationality. All responses were reviewed by the authors without coding in order to collaboratively identify common themes. Initial themes were distinguished based on whether participants commented on the strength versus continued vulnerability of Indigenous peoples. One of the authors (NB) coded all of the data, revising and extending codes as responses were reread, creating a continuous retro-deductive process [26]. Reponses could be coded in more than one category, but could only be coded once for any given theme. The remaining authors reviewed the coding, and differences were resolved by consensus.

As can be seen in Table 3, when the scenario ended on the discourse of *vulnerability*, Canadians were particularly likely to acknowledge the **victimization** of Indigenous peoples as a result of the IRSs. This included statements indicating that "cultural trauma like that can stay in a community for a long time, and has a lot of detriment to its people", and that "these people are now damaged because of the abuse". The salience of the victimization of Indigenous peoples tended to co-occur with focusing on what had been done **in the past** (e.g., "these sins were prevalent decades ago"), $X^2(1) = 5.15$, $p = .023$. When the scenario highlighted the *strengths* of Indigenous peoples, Americans were especially likely to describe their **resilience** and capacity to persevere in the face of adversity (e.g., ". . . they have had to endure and fight hard for their rights."). In addition, among Americans, the strength discourse evoked a greater sense that Indigenous peoples should have benefited by **learning** from the experience (e.g., ". . .to value all cultures of people and learn how to take what is given to them and use it effectively"). This said, Canadians appeared more likely to raise the lessons learned when the scenario emphasized the vulnerability of Indigenous peoples, and these lessons tended to reflect a learned "mistrust" and "resentment of outsiders". Although both American and Canadian participants raised the issue of Indigenous peoples being **entitled** to compensation or reparations, this was done both in the affirmative, and through explicit statements that there should be ". . . [no] special privileges because it undermines the need to have a sense of choice and willpower

**Table 3. Frequencies (%) of participants in each discourse condition (Vulnerable vs. Strong) as a function of nationality (American vs. Canadian) expressing each theme in written meaning making task in Study 2.**

| | American | | | Canadian | | |
|---|---|---|---|---|---|---|
| | Vulnerable | Strong | $X^2(1)$ | Vulnerable | Strong | $X^2(1)$ |
| Victimized | 15 (39.5%) | 9 (30.0%) | 0.66 | **21 (67.7%)** | 12 (34.3%) | 7.36** |
| In the Past | 15 (39.5%) | 7 (20.0%) | 3.28 | **19 (57.6%)** | 12 (33.3%) | 4.09* |
| Resilient | 5 (13.2%) | **27 (77.1%)** | 30.30*** | 3 (9.1%) | 14 (38.9%) | 8.23** |
| Learning | 9 (23.7%) | **19 (54.3%)** | 7.22** | **12 (36.4%)** | 5 (14.3%) | 4.62* |
| Entitlement | 16 (47.1%) | 16 (57.1%) | 0.90 | 12 (40.0%) | 14 (41.4%) | 0.01 |

* $p < .05$;

** $p < .01$;

*** $p < .001$.

instead of having everything handed to you just because something unfair happened to your ancestors". Such considerations did not vary as a function of the scenario discourse. In short, the qualitative responses were consistent with nationality differences in perceptions of Indigenous peoples as benefiting from the IRS experience, such that Americans were more likely to be sensitive to a discourse focusing on strengths, and although Canadians acknowledged victim suffering, they were more likely to express views that served to diminish ingroup responsibility.

## Discussion

When the victimization of Indigenous peoples in the IRSs was made salient, as expected, non-Indigenous Canadians and Americans differed in their responses to the discourse employed to describe Indigenous strengths versus vulnerabilities. Consistent with previous research conducted with third-party observers [4], Americans were more likely to perceive greater benefit finding when Indigenous peoples were depicted as strong and as overcoming their past suffering, in comparison to when they were portrayed as vulnerable. By portraying Indigenous peoples as having persevered despite the adversities encountered, third-party observers' perceptions that they were able to find strength and grow from the experience were not entirely unreasonable [23]. Indeed, based on the qualitative analysis of their meaning-making reflections, Americans were especially likely to acknowledge the resilience of Indigenous peoples and that they had benefited by learning to value their own culture and traditions.

As in Study 1, non-Indigenous Canadians' perceptions of benefits derived from the IRS experience and expectations regarding Indigenous peoples' moral actions were solely influenced by the extent to which they endorsed modern racist attitudes, and the role of racism was especially strong when they regarded Indigenous peoples as continuing to experience discrimination. The central role of racism might suggest that non-Indigenous Canadians' reactions to Indigenous peoples may constitute an effort to affirm their own group identity by justifying the benefits that Indigenous peoples ought to derive from their experience [13], especially when the harm perpetrated against them was viewed as ongoing [15]. However, the finding that this mediated relationship was strongest when victims continued to experience discrimination also held for non-Indigenous Americans, which suggests that racism in itself serves to strengthen motivated biases to justify ongoing social inequities [27].

At the same time, the qualitative analyses suggest that when the vulnerability of Indigenous peoples was salient, non-Indigenous Canadians were more likely to acknowledge the harm done, and that the perpetration of the IRSs resulted not so much in deriving strengths, but in learning to mistrust the motives of others. While these perceptions might intuitively foster greater empathy and understanding, consistent with a just world perspective, highlighting the vulnerability of Indigenous peoples did not reduce participants' propensity to diminish perceptions of benefits found or attribute lower moral obligations [10, 11]. Particularly under these conditions, non-Indigenous Canadians may have diminished their responsibility for making amends by regarding the harms done as belonging in the past, and that efforts to reconcile differences are unlikely to succeed in light of Indigenous peoples' learned mistrust.

## General discussion

The aim of the present investigation was to assess non-Indigenous Canadians' moral expectations of Indigenous peoples, given the history of collective victimization at the hands of the Canadian government and religious institutions. Past research had demonstrated that historically victimized groups tend to be held to a higher standard of conduct when the meaning of the victimization for the victims is reflected on, by virtue of being perceived to benefit by

learning from their suffering, and hence to be better for it [2–5]. At the outset of this investigation, it was uncertain whether patterns of perceptions of moral obligation would differ from past research that has focused on third-party observers, given Canadians' historical perpetrator status.

Indeed, non-Indigenous Canadians were less likely to regard Indigenous peoples to have derived psychological benefits from the IRS experience, or to perceive higher moral obligation in comparison to the views of Americans. In addition, past research has found that such perceptions were exacerbated when observers focused on the meaning of the suffering for the victims, rather than on the lessons learned by the perpetrator group [3, 5]. In the present investigation, the possibility was raised that focusing on implications for the perpetrator group (to which participants belonged), a defensive response would be elicited among non-Indigenous Canadians [13, 14]. Perhaps for this reason, in Study 1, there were no differences in perceptions of the benefits and obligations of Indigenous peoples as a function of whose experience participants focused on (victim, perpetrator, or control).

As in past research, across both studies in the present investigation, a positive relation was found between perceptions of the victim group gaining psychological benefits from their experience and expectations of their moral obligation to others. It was additionally shown that such perceptions were greater among those who were more likely to express modern racist views. Given that non-Indigenous Canadians' own group was the historical perpetrator of the harm done to Indigenous peoples in the IRS system, racism was proposed to motivate the need to defend a positive collective identity by deriving victim benefits and moral obligations. However, racism was also associated with such perceptions among non-Indigenous Americans, who were third-party observers of the IRS legacy. This suggests that the benefits found and obligations attributed to victims were not simply an effort to affirm the moral righteousness of one's own group grounded in modern racist views, but likely represents a ubiquitous strategy for re-asserting just world beliefs that may be a pervasive form of system justification [27]. Nonetheless, oppressive policies pertaining to Indigenous peoples are as extensive in the United States as they are in Canada. Thus, it is possible that even though the present study focused on the Canadian experience, Americans' awareness of comparable practices in their own country might have indirectly resulted in identifying with the Canadian situation (or their common identification as settlers), resulting in a similar response pattern. Such identification was not assessed in the present study.

Although the pattern of relations among modern racism, perceived Indigenous benefit-finding and moral obligation was consistent across studies, it was also found that greater racism was associated with lower perceptions that Indigenous peoples continued to experience discrimination. Such a relation was not surprising given that beliefs that the victim group should "not complain", "get on with their lives", and be treated "just like everyone else" are inherent to modern racism [17]. Indeed, when participants were unable to minimize perceptions of the current victimization of Indigenous peoples, the relations between racism and perceptions of benefits found and moral obligation were exacerbated. This, too, occurred irrespective of national group. Although this pattern of findings is correlational, it suggests that a group's continued experience of victimization may trigger reactions that motivate observers to regain a sense that those who suffer have been justly compensated (belief in a just world) [3], and is consistent with research demonstrating that more impactful victim experiences exacerbate the tendency to derogate the victim [10].

Highlighting victimization not only triggered processes associated with modern racism, it further appeared to reduce historical perpetrator group members' sensitivity to contextual cues that conveyed victim group strengths. In particular, non-Indigenous Canadians were less influenced than (non-racist) Americans by the framing of Indigenous peoples as strong or

vulnerable. Moreover, while Americans were responsive to the strengths of Indigenous peoples, picking up on their resilience and positive values learned, Canadian participants were more likely to derive meaning for victims that highlighted Indigenous peoples' past victimization, and negative lessons associated with not being able to trust others. In effect, non-Indigenous Canadians appeared to view Indigenous peoples as an underdog that continues to struggle against the odds. While such a perception might ordinarily elicit sympathy [28], support for the underdog does not translate into practical solidarity that achieves change, especially if positive outcomes compete with the interests of the perceiver, or if the outcomes have broad societal consequences [28]. If Indigenous peoples' efforts to achieve social justice compete with the interests of non-Indigenous Canadians, they might have their sympathy but not their support [12, 29]. Indeed, Indigenous efforts to achieve equality and compensation for historical injustices, and their active exercise of treaty rights, continue to be opposed by many non-Indigenous Canadians [30].

Although framing the status of the victim group as strong versus vulnerable did not appear to alter non-Indigenous Canadians' propensity to bestow benefits and apply moral expectations of Indigenous peoples, it is possible that members of the victim group itself might benefit from the salience of their strengths in spite of these past experiences [23, 24]. The present studies did not assess the views of Indigenous peoples, but others have suggested that the validation of past victimization is self-affirming, provides meaning to current conditions [31–33], and can improve victims' well-being as they are able to acknowledge their perseverance and resilience [24, 34]. In this regard, Indigenous peoples are increasingly shaping the public narrative to contextualize their current challenges through the lens of their past experiences, but are doing so from a position of strength—resistance, reclaiming, and renewal.

To the extent that Indigenous peoples are successful in conveying their strengths and their inherent right to equity, respect and self-determination, any violation of the moral expectations placed upon them could come at a high social cost [3, 35]. Specifically, although victim blame was not assessed, highlighting the strengths of Indigenous peoples could potentially lead to the belief that the conditions they have faced and continue to face are of their own doing. Indeed, such victim-blame rhetoric has been evident as Indigenous peoples demand that the disproportionate number of missing and murdered Indigenous women and girls be addressed. Public and political attention gets predictably diverted, without contextualization, to the domestic abuse experienced by Indigenous women at the hands of 'familiar' others [36]. Likewise, consistent with modern racist views that Indigenous peoples should be satisfied with what they have and that any additional demands are looking for special treatment, a focus on strengths may undermine non-Indigenous Canadians' perceptions that further change is needed. Given the challenges associated with highlighting either past (and continued) victimization of Indigenous peoples, as well as with promoting a strength-based narrative, finding an effective balance that promotes respect, affirmation, and social change is an ongoing debate among Indigenous advocates and leaders [24].

Given the complex motivational biases and intergroup dynamics associated with historical victimization, the implications for Indigenous peoples as they grow in strength and continue to fight to address inequities and achieve self-determination, are challenging. The results of the present study suggest that such efforts are likely to mobilize opposition from those who believe that such inequities have already been sufficiently addressed, and that Indigenous peoples should 'get over' their past (modern racism). The finding that modern racist beliefs were associated with perceiving Indigenous peoples to have benefited or learned from their past victimization and, as a result, they should be better as a people were evident irrespective of the relationship held with the victim group, and were especially pronounced when the victim group continued to encounter systemic discrimination. However, unlike third-party observers

(Americans) who differentially regarded Indigenous peoples as a function of whether their strengths versus continued vulnerabilities were made salient, this was not the case among non-Indigenous Canadians. Although the Canadian government has committed to healing and reconciliation between Indigenous peoples and Canada, such a relationship is unlikely to be achieved if addressing past wrongs is more likely to elicit psychological processes that serve the perpetrator group's need to believe that justice has already been served.

Although the findings of the present research are not especially optimistic, there are limitations to their generalizability. Participants were not a representative sample of the non-Indigenous Canadian population, but rather comprised students and online survey respondents, who were relatively informed of the issues associated with the IRS system and reported fairly low levels of modern racism. Perhaps for this reason, our efforts to manipulate the salience of particular features of the implications of the IRS experience for Indigenous peoples was limited, and may have been more evident with a more representative sample. Moreover, our key predictors were correlational (modern racism) or quasi-experimental (nationality). Thus, other co-occurring factors might account for the relationships with perceptions of victim benefit finding and moral expectations, such as variations in contact with Indigenous peoples. Greater positive intergroup contact is associated with less prejudicial attitudes and greater support for social change [37]. Such predictors were not assessed in the present investigation, and so their role is uncertain. However, they point to opportunities to foster mutual understanding, and alter the relationship between Indigenous peoples and non-Indigenous Canadians. In particular, the findings of the present studies suggest that mobilizing allies who do not espouse modern racist perspectives to work together with Indigenous peoples might enable the balanced narratives and nuanced understandings needed to bring about social equity.

## Author Contributions

**Conceptualization:** Mackenzie J. Doiron, Nyla Branscombe, Kimberly Matheson.

**Formal analysis:** Mackenzie J. Doiron, Nyla Branscombe, Kimberly Matheson.

**Funding acquisition:** Nyla Branscombe, Kimberly Matheson.

**Methodology:** Nyla Branscombe, Kimberly Matheson.

**Project administration:** Mackenzie J. Doiron, Kimberly Matheson.

**Supervision:** Kimberly Matheson.

**Writing – original draft:** Mackenzie J. Doiron.

**Writing – review & editing:** Nyla Branscombe, Kimberly Matheson.

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
