## [Decision Letter · Decision Letter 0]

9 Mar 2021

PONE-D-20-31541

Canadians’ moral expectations of Indigenous peoples in light of the salience of past victimization

PLOS ONE

Dear Dr. Matheson,

Thank you for submitting your manuscript to PLOS ONE. After careful consideration, we feel that it has merit but does not fully meet PLOS ONE’s publication criteria as it currently stands. Therefore, we invite you to submit a revised version of the manuscript that addresses the points raised during the review process.

Please review and respond to each of the comments from the reviewers in your updated submission. We look forward to receiving the revised version.

We look forward to receiving your revised manuscript.

Kind regards,

Amy Michelle DeBaets, PhD

Academic Editor

PLOS ONE

Journal Requirements:

2. Please provide additional details regarding participant consent.

In the ethics statement in the Methods and online submission information, please ensure that you have specified what type you obtained (for instance, written or verbal, and if verbal, how it was documented and witnessed).

If your study included minors, state whether you obtained consent from parents or guardians.

If the need for consent was waived by the ethics committee, please include this information.

3. We note that you have stated that you will provide repository information for your data at acceptance. Should your manuscript be accepted for publication, we will hold it until you provide the relevant accession numbers or DOIs necessary to access your data. If you wish to make changes to your Data Availability statement, please describe these changes in your cover letter and we will update your Data Availability statement to reflect the information you provide

Reviewers' comments:

Reviewer's Responses to Questions

**Comments to the Author**

1. Is the manuscript technically sound, and do the data support the conclusions?

Reviewer #1: Yes

Reviewer #2: Yes

2. Has the statistical analysis been performed appropriately and rigorously? 

Reviewer #1: I Don't Know

Reviewer #2: Yes

3. Have the authors made all data underlying the findings in their manuscript fully available?

Reviewer #1: Yes

Reviewer #2: Yes

4. Is the manuscript presented in an intelligible fashion and written in standard English?

Reviewer #1: Yes

Reviewer #2: Yes

5. Review Comments to the Author

Reviewer #1: I have been asked to review this manuscript because of my expertise in Residential School history and the Canadian Truth and Reconciliation Commission process and findings. Related to this, your representation of the process is basically sound. I would suggest the following minor issues be addressed before publication:

Line 46 ...perpetrated by the Canadian government *and Christian churches (both were party to the settlement agreement)

Line 66 *official residential school partnership between government and churches was late 1800s

Line 69 *be careful with numbers here. The TRC had records that indicated 4200 children died at residential school. The National Centre for Truth and Reconciliation continues to uncover unreported missing children and estimate the number is significantly higher. They have researched and found the names of 2800 of those children (commemorated in September 2019). If you choose to name the 6000 estimate, you should include the page number that cites this in the report.

Line 86 *consider ‘systemically disadvantaged; rather than disadvantaged, which has complicated implications

Line 89 *consider modifying or expanding ‘victimization’ the first time you use it in this section as many Indigenous people reject this characterization. I realize this is a part of your research terms. Perhaps victims or recipients of cultural genocide.

Line 288 *terminology in the United States is 'Native Americans'. Consider this as an adequate substitute for American Indians.

With these modifications, and with the affirmation of those in your direct field, your research makes an excellent contribution to Canadian understanding of post-TRC perceptions.

Reviewer #2: Dear Editors,

I have now had a chance to read and review the authors paper “Canadians’ moral expectations of Indigenous peoples in light of the salience of past victimization”. My apologies for the small delay in submitting my review. In general, the manuscript is well-structured and the two experimental studies are well-designed and well-executed. Their findings overall are compelling and represent an important contribution to not only our developing understanding of important psychological factors affecting how perpetrator and third party observer groups respond to different representations of victimized groups, but more specifically they have real implications for ongoing discourse and action in the context of Canadian attempts at reconciliation and healing with Indigenous peoples.

I recommend that the manuscript be accepted with some minor revisions. Below, I detail a list of observations, comments and suggestions for the authors to review.

A central recommendation involves improving some clarity of concepts especially early on in the manuscript: Key terms “third party observers” “benefit finding”, “meaning making” and “higher standard of moral behavior/obligation” should be defined more clearly earlier on in the paper, ideally with reference to a clear contemporary or historic example from the Canadian context. This would generally help improve the clarity of the authors’ key ideas from the start. More consistency of terms (e.g. “non-Indigenous Canadians” rather than “Canadians” throughout) would also help improve precision.

More generally, I encourage the authors to attend closely to my comments on some of the framing of their discussion. As I know the authors well-understand, this is a highly complex context to do research within. While their findings are compelling, I encourage the authors to consider whether they truly warrant making such grim predictions for the likely success of Indigenous efforts to achieve justice, reconciliation and healing within the Canadian context. The authors find evidence consistent with barriers to (some) members of perpetrator groups and third party observers appropriately responding to evidence of a group’s historical victimization in ways that may lead to justice or reparation. Rather, they may justify the system that benefits them. This is does not preclude the possibility of other avenues to social change or justice, especially ones led by Indigenous peoples themselves. Some more nuance around the implications of the authors findings exists within their own data (e.g. the content analysis) and the wider literature, and could be referenced in order to bring out more of this complexity to offer a slightly less hopeless conclusion.

In general though, I was impressed by the authors’ careful work and am pleased to see more of this kind of research happening in this context. I wish the authors all the best in their revisions and look forward to seeing this manuscript in published form soon.

More detailed comments:

The title could perhaps be revised to be more reflective of the fact that the study focused on Non-Indigenous Canadians and Americans, not simply or exclusively “Canadians”.

p. 3 line 52 Tense might make more sense in the present, e.g. “expect” not “expected” and in the following line suggest: “…members of victimized groups are often expected to be better, stronger people, and are held to a…”

p. 3 lines 50-55 - Clarity of why this matters for the issue at hand could be improved. If the purpose is for the Canadian public to accept the TRC calls to action, the authors have not made it clear yet why it should matter that third party groups (a definition might be necessary before the reader can understand who this refers to and why they matter for this issue - indeed “third party observers” is never adequately defined in this paper and should be) expect victim groups to have “learned from the experience”. This has no obvious bearing on whether or not the Canadian public will “accept” TRC calls to action. Suggest reframing this paragraph so it is immediately clear to readers how these past studies connect to the problem as presented and how these points are related. I would also encourage the authors to make it very clear why we should care about third party observers at all - what is their relevance for the kinds of policy problems (e.g. Canadians not accepting or resisting TRC calls to action) the authors have in view?

p. 3 line 56 - this line is a bit unclear and could be reworded. I initially read it as Canadians having “represented” (as in the act of developing social representations) historical perpetrators but then realized the authors meant “Canadians” simply ARE the historical perpetrators. This should be clarified. I would also suggest more specific intergroup language such as “Settlers” or “non-Indigenous Canadians” as Indigenous peoples in Canada are arguably still Canadians (as the authors do later in the paragraph). The language could be more specific and consistent here to help improve clarity for readers.

p. 3 line 58 - “similar reaction” is referring to the research briefly reviewed in the preceding paragraph but that connection is not immediately apparent and could be better connected/specified. In general the first paragraph could be laid out more clearly so the connection to the present study is more direct.

p. 5 line 96-97 - the connection between belief in a just world and victimized groups having benefitted/learned from their victimization is not apparent and could be spelled out in more detail. This becomes more clear by the end of the paragraph, but perhaps could be introduced more clearly.

p. 6 lines 126-130 is very nice and clearly described, especially in terms of who exactly is the historical perpetrator. Perhaps make the above introduction more like this. (p. 3 line 56)

p. 6 line 28-30 - More generally, this is an intriguing possibility but feels like it could have benefitted in the intro from more of a contextual set up. For example, is there anecdotal evidence of this happening in public discourse within Canada about IRS? Senator Lynn Beyak’s infamous letters/intransigence comes to mind and could serve as an example illustration earlier on in the paper perhaps?

p. 7 line 136 this “search for meaning” in IRSs comes out of nowhere a bit and could be better contextualized or defined beforehand. In what sense might non-Indigenous Canadians’ awareness of the IRS system trigger a “search for meaning”? This is not clear, and needs to better explained. I see that a “search for meaning” is defined more specifically in the next section (Study 1 intro description) and I do not think “search for meaning” broadly captures what is actually meant.

p. 7 line 144-145 This line is awkward eg.. “to perceive a victim to find” should be reworded. This is also the only time in the paper the adjective “prosocial” is used with “behaviour” so I suggest removing this as it is not central to your explanations or analysis and of course has its own entire literature, which you are not referencing. Perhaps just using ‘moral behaviour” consistently is more precise? Or “a higher standard of morality”?

p. 7 line 152-154 - It strikes me that some of this explanation of past research is very abstract and difficult to understand or apply to this particular context without a concrete example. What would it look like for “third party observers” (still not very clearly defined who this would be…) to engage in a meaning making process around IRS and then conclude that IRS benefitted Indigenous people and Indigenous people are now held to a higher moral standard. Are there contemporary or historic examples to illustrate this? Perhaps adding one in earlier on (in the context of the very nice and clear description of the history of the IRS system?) would improve clarity throughout this intro? Once again Senator Lynn Beyak’s recent story comes to mind as a possibility to highlight?

p. 8 - Great to see a thoughtful power analysis!

p. 9 line 185-187 - this is the most clearly I have understood what has been meant by “meaning making” previously. Perhaps you can more clearly explain you are simply asking people what was “learned” or what the “implications” were related to IRS for different groups? That may make your research questions more clear earlier on. Perhaps you could frame the intro more as something like “Learning about the IRS system for the first time has set off a process of reflection and wider meaning-making for non-Indigenous Canadians which may differ depending on several factors including if they are thinking about the implications/lesons for non-Indigenous versus Indigenous Canadians.” Or something like that. If this is a clear summary of the point of these studies it was not as clear as it should be earlier on.

Page 10 lines 205-206 “meaning making condition did not significantly affect levels of racism reported” could be rephrased more clearly.

P. 11 a figure depicting the mediation model would have been useful

p. 12 line 256-257 - This is a plausible explanation for the different effects here and I’m glad the authors raise the importance of participants being members of the historical perpetrator group

P. 12 lines 260-262 - this line suggests there actually WERE benefits bestowed via IRS…suggest change to “…through the benefits some participants may have believed were bestowed by the IRS system” to make this more clear, and less problematic.

p. 12. Line 262 “the Indigenous peoples” should just be “Indigenous peoples” or perhaps “Indigenous peoples in Canada” to be consistent and precise

p. 13 line 265 I note you speak of just “observers” rather than “third party observers” here - perhaps the “third party” is unnecessary and could be removed elsewhere too?

P. 13 - the rationale for Study 2 is beautifully described! Such an important issue that has not received enough attention in the discourse, scholarly or otherwise. I don’t mean to toot my own horn here but it bears mentioning that some of my own work is at least somewhat relevant to this study in particular. It is a qualitative study reporting Indigenous peoples’ own preferences for different representations of colonial history and we unpack some of the implications of more strength vs deficit based framings for Indigenous-settler intergroup relations in our discussion section: https://jspp.psychopen.eu/index.php/jspp/article/view/5185/5185.pdf We also (I believe…) encouraged more experimental work on this topic - which the authors are somewhat responding to with this study. To be clear, I am Not suggesting the authors need to cite this - but it may be a useful source to consult as the authors continue their work on this topic. It at least suggests that it is not only the media contributing to this discourse but Indigenous people themselves as well.

P. 13-14 - I think it’s brilliant to contrast the Canadian vs. American non-Indigenous samples in this study. Nicely done.

P. 15 - I also thought this was an excellent manipulation. As above, this work is very important. Very interesting to see how these discourses are functioning in different ways between Canadians and Americans.

P. 19 Wonderful to see this content analysis of the differential American/Canadian responses to the manipulation primes. This is an effective and compelling approach to treating qualitative data of this nature and considerably adds to our understanding of the overall story here. The X2 analysis and summary table bring this all together very effectively. They highlight for us the importance of recognizing that certain categories the authors had operationalized in the quantitative studies (e.g. “benefit finding”) can be understood in different and more nuanced ways, e.g. a more paternalistic “they should have learned to not make others suffer” vs. a more political / resistant “they learned not to trust outsiders”. Some discussion of how these findings may nuance prior qualitative findings and operationalizations of these concepts in the experimental components of the study would have been a welcome addition.

Regarding coding, I did wonder if the authors could at least provide a rough picture of how long these responses were on average (character count average and range could work) and if they could at least tell the reader how many people coded these and if there was a process for resolving disagreement over two coders who differed on a specific code.

Finally, more could likely be made of this qualitative analysis though and perhaps the authors will consider a more interpretive analysis of these qualitative data on their own at some point, in another paper.

p. 21 - as in the intro, it is important to distinguish between “Indigenous peoples” and “non-Indigenous Canadians”. The authors sometimes use the word “Canadians” to refer to “non-Indigenous Canadians” in their sample when it would be more appropriate to say non-Indigenous Canadians consistently as exclusion criteria specifically ensured no Indigenous Canadians were present in the sample. The implication here (which I don’t think the authors intend) that “Canadians” does not include Indigenous peoples is potentially problematic and of course has a long and complex political history. For example in line 447 it should say “…non-Indigenous Canadians’ reactions to Indigenous peoples…” (as in a number of other places…)

p. 22 line 465 should read “non-Indigenous Canadians” - the authors should do a thorough check for other instances of unqualified “Canadians” that should specify “non-Indigenous Canadians”, same for line 480

p. 23 lines 495-496 the authors distinguish between “perpetrators” and “third party observers” - calling back into question how clearly these different groups were defined or differentiated in the intro set up. The intro to the paper would at times seem to equate these groups, or at least focus specifically on “third party observers” when both groups are clearly differentiated and different focuses of the paper.

p. 24 the general discussion presents powerful implications and arguments that highlight the importance of the authors’ findings in a compelling and clear way.

p. 24 line 515 - I’m not sure saying that the different framings (strong vs vulnerable) is best described as having been “lost on perpetrator group members” - this suggests a more general indifference than the data warrant and could be phrased more specifically: these frames did not seem to affect what we measured. This does not rule out other implications those framings could have that are beyond the scope of what this study examined.

p. 25 As mentioned above - this may be a worthwhile moment to consider acknowledging that Indigenous communities themselves engage in internal debate about how to represent themselves and the impact of colonial harms and the various pros and cons of different representations (e.g. victims vs resilient) for intergroup relations. Our qualitative work examining this in the context of an urban Indigenous community may be worth referencing here: https://jspp.psychopen.eu/index.php/jspp/article/view/5185/5185.pdf Once again, this is obviously not a requirement and I leave it to the authors’ discretion.

P. 25 line 525 - I am slightly unclear how highlighting Indigenous strengths may lead to outgroups blaming them for having created the adverse conditions Indigenous people continue to face…perhaps this could be clarified? However, on a related note and In part based on our work on this topic, I would encourage the authors to consider that there are other potential costs and benefits to emphasizing the success/resilience of Indigenous communities beyond the suggestion by outgroup members that they are “to blame” for what has happened to them. The impacts of this representation also may vary depending on their audience (Indigenous vs. non-Indigenous). For example resilience and resistance stories shared amongst Indigenous community members are a valuable source of in-group pride, but on the other hand may weaken claims for ongoing support from perpetrator outgroups: if you’re so resilient why do you still need our help? Likewise for representations of the “damage” inflicted in Indigenous communities via colonial policy. These representations can be drawn on as an excuse within Indigenous communities for certain negative behaviours, but alternatively these representations can be identity-protective in that they may help some Indigenous people better explain to themselves and other community members/family “why things are the way they are” in ways that attribute harms to external (what they did to us) rather than internal (who we are) factors. These possibilities may have implications for other parts of the authors’ findings. They at least suggest that there are more factors to consider than what outgroups will think about Indigenous peoples if they learn that they are resilient and strong.

p. 25 line 531-533 - the line “The finding that modern racist beliefs were associated with perceiving Indigenous peoples to have benefited or learned from their past victimization and, as a result, they should be better as a people…” could perhaps be more nuanced to take into account what the authors found in the content analysis: sometimes participants felt that what Indigenous peoples may have “learned” from their experiences of victimization were self-protective norms of mistrust of outsiders etc. - this is not quite consistent with the more simplified notion I fear the authors occasionally imply (and that is implied here) that “learning from the experience” is the same as “benefitting from the experience”. Is someone who says “Indigenous people learned to become more vigilant and mistrustful of settlers as a result of the IRS system” saying that they “benefitted” from these schools in the same way senator Lynn Beyak has said Indigenous people “benefitted” from the IRS system? (https://www.cbc.ca/news/politics/senator-lynn-beyak-suffered-residential-schools-1.4042627) I would think not - the authors could perhaps do more to clarify the differences here in how participants were interpreting and responding to their “benefit finding” measure, as the content analysis of written responses suggested.

P. 25 - does this study have any limitations? None are noted. Issues of generalization at least are important to note, particularly if the authors wish to generalize from their findings to make rather grave pronouncements of the likely failure of reconciliation or healing in Canada…see below

p. 25 lines 539-541: The line “…such a relationship is unlikely to be achieved if addressing past wrongs is more likely to elicit psychological processes that serve the perpetrator group’s need to believe that justice has already been served.” Is a rather grim way to end the paper…are there no implications or findings implied by the paper that suggest a remedy? I encourage the authors to at least find a glimmer of hope to offer as their parting statement!

6. PLOS authors have the option to publish the peer review history of their article (what does this mean?). If published, this will include your full peer review and any attached files.

Reviewer #1: No

Reviewer #2: **Yes: **Scott Neufeld

---

## [Author Response · Author response to Decision Letter 0]

4 Apr 2021

Response to reviewers

Reviewer #1 

Line 46 ...perpetrated by the Canadian government *and Christian churches (both were party to the settlement agreement)

The role of the church has been added (line 45-46).

Line 66 *official residential school partnership between government and churches was late 1800s

This has been clarified (line 71)

Line 69 *be careful with numbers here. The TRC had records that indicated 4200 children died at residential school. The National Centre for Truth and Reconciliation continues to uncover unreported missing children and estimate the number is significantly higher. They have researched and found the names of 2800 of those children (commemorated in September 2019). If you choose to name the 6000 estimate, you should include the page number that cites this in the report.

We have rephrased as follows (line 74-75):

an estimated 4,200 would never return home alive (although it is believed that the number is significantly higher)….

Line 86 *consider ‘systemically disadvantaged; rather than disadvantaged, which has complicated implications

We have made this change (line 92).

Line 89 *consider modifying or expanding ‘victimization’ the first time you use it in this section as many Indigenous people reject this characterization. I realize this is a part of your research terms. Perhaps victims or recipients of cultural genocide.

We agree that many Indigenous people reject the ‘victim’ characterization. We have therefore changed the terminology to ‘harms perpetrated against Indigenous peoples’ (line 96). Thereafter, given the description of harms perpetrated against Indigenous peoples in Canada described in lines 71-91, the context of the term ‘victimization’ has been provided. As the reviewer notes, to connect the terminology of the theoretical framework to the experience of Indigenous peoples, the use of the term ‘victim’ is important. This said, recognizing that Indigenous peoples are indeed not ‘passive victims’, but have demonstrated strength and resistance, even as systemic discrimination continues was a primary purpose of Study 2.

Line 288 *terminology in the United States is 'Native Americans'. Consider this as an adequate substitute for American Indians.

Our terminology has been corrected (line 314) 

Reviewer #2

A central recommendation involves improving some clarity of concepts especially early on in the manuscript: Key terms “third party observers” “benefit finding”, “meaning making” and “higher standard of moral behavior/obligation” should be defined more clearly earlier on in the paper, ideally with reference to a clear contemporary or historic example from the Canadian context. This would generally help improve the clarity of the authors’ key ideas from the start. More consistency of terms (e.g. “non-Indigenous Canadians” rather than “Canadians” throughout) would also help improve precision.

More generally, I encourage the authors to attend closely to my comments on some of the framing of their discussion. As I know the authors well-understand, this is a highly complex context to do research within. While their findings are compelling, I encourage the authors to consider whether they truly warrant making such grim predictions for the likely success of Indigenous efforts to achieve justice, reconciliation and healing within the Canadian context. The authors find evidence consistent with barriers to (some) members of perpetrator groups and third party observers appropriately responding to evidence of a group’s historical victimization in ways that may lead to justice or reparation. Rather, they may justify the system that benefits them. This is does not preclude the possibility of other avenues to social change or justice, especially ones led by Indigenous peoples themselves. Some more nuance around the implications of the authors findings exists within their own data (e.g. the content analysis) and the wider literature, and could be referenced in order to bring out more of this complexity to offer a slightly less hopeless conclusion.

Reviewer #2s general comments are addressed in response to the more detailed comments below.

More detailed comments:

The title could perhaps be revised to be more reflective of the fact that the study focused on Non-Indigenous Canadians and Americans, not simply or exclusively “Canadians”.

The reviewer’s concerns about our unqualified use of the term ‘Canadians’ is recognized, and we’ve addressed this throughout the manuscript, including the title. The exception is in reference to the meaning-making manipulation in Study 1. In this instance, participants were directed to consider the implication of the IRSs for ‘Canadians’.

p. 3 line 52 Tense might make more sense in the present, e.g. “expect” not “expected” and in the following line suggest: “…members of victimized groups are often expected to be better, stronger people, and are held to a…”

This modification has been made (lines 52-54).

p. 3 lines 50-55 - Clarity of why this matters for the issue at hand could be improved. If the purpose is for the Canadian public to accept the TRC calls to action, the authors have not made it clear yet why it should matter that third party groups (a definition might be necessary before the reader can understand who this refers to and why they matter for this issue - indeed “third party observers” is never adequately defined in this paper and should be) expect victim groups to have “learned from the experience”. This has no obvious bearing on whether or not the Canadian public will “accept” TRC calls to action. Suggest reframing this paragraph so it is immediately clear to readers how these past studies connect to the problem as presented and how these points are related. I would also encourage the authors to make it very clear why we should care about third party observers at all - what is their relevance for the kinds of policy problems (e.g. Canadians not accepting or resisting TRC calls to action) the authors have in view?

Our interest was not in the reactions of third-party observers – however, this is who the preponderance of research conducted in this area has focused on. This said, the distinction between observers (not the victims) and third-party observers (not the victims, nor connected historically or presently to the perpetrators) is not important until we describe the rationale for the present investigation (i.e., why we might question whether non-Indigenous Canadians’ perceptions may differ from past research). Thus, the term ‘third-party’ has been removed until it is appropriate, and we define it at this point (lines 121-123). Thereafter, it is only used in reference to making the distinction between non-Indigenous Canadians and Americans.

p. 3 line 56 - this line is a bit unclear and could be reworded. I initially read it as Canadians having “represented” (as in the act of developing social representations) historical perpetrators but then realized the authors meant “Canadians” simply ARE the historical perpetrators. This should be clarified. 

This sentence has been removed as part of another revision suggested by Reviewer #2.

I would also suggest more specific intergroup language such as “Settlers” or “non-Indigenous Canadians” as Indigenous peoples in Canada are arguably still Canadians (as the authors do later in the paragraph). The language could be more specific and consistent here to help improve clarity for readers.

As noted earlier, we have clarified this distinction throughout.

p. 3 line 58 - “similar reaction” is referring to the research briefly reviewed in the preceding paragraph but that connection is not immediately apparent and could be better connected/specified. In general the first paragraph could be laid out more clearly so the connection to the present study is more direct.

We considerably revised this paragraph delineating the goals of the present study, and hope that it is now much clearer (lines 55-69)

p. 5 line 96-97 - the connection between belief in a just world and victimized groups having benefitted/learned from their victimization is not apparent and could be spelled out in more detail. This becomes more clear by the end of the paragraph, but perhaps could be introduced more clearly.

We have added a more detailed rationale and reorganized this paragraph to clarify the importance of people’s need to believe in a just world in accounting for why they would perceive benefit in the victim’s suffering and hold them morally accountable (lines 103-120).

p. 6 lines 126-130 is very nice and clearly described, especially in terms of who exactly is the historical perpetrator. Perhaps make the above introduction more like this. (p. 3 line 56) 

As a result of this suggestion, we have revised this introductory paragraph, as noted above (lines 55-69)

p. 6 line 28-30 - More generally, this is an intriguing possibility but feels like it could have benefitted in the intro from more of a contextual set up. For example, is there anecdotal evidence of this happening in public discourse within Canada about IRS? Senator Lynn Beyak’s infamous letters/intransigence comes to mind and could serve as an example illustration earlier on in the paper perhaps?

Thank you for raising the Lynn Beyak example in the context of this study. We have now incorporated it as an example of how some non-Indigenous Canadians sought to identify benefits as a result the IRS experience (lines 136-140). 

p. 7 line 136 this “search for meaning” in IRSs comes out of nowhere a bit and could be better contextualized or defined beforehand. In what sense might non-Indigenous Canadians’ awareness of the IRS system trigger a “search for meaning”? This is not clear, and needs to better explained. I see that a “search for meaning” is defined more specifically in the next section (Study 1 intro description) and I do not think “search for meaning” broadly captures what is actually meant.

We have now anticipated this meaning-making reflection process more explicitly earlier in the introduction (e.g., lines 61-64; 151-153). In addition, we have reorganized the first paragraph of Study 1 to better lead into the rationale for our meaning-making manipulation (lines 162-173).

p. 7 line 144-145 This line is awkward eg.. “to perceive a victim to find” should be reworded. This is also the only time in the paper the adjective “prosocial” is used with “behaviour” so I suggest removing this as it is not central to your explanations or analysis and of course has its own entire literature, which you are not referencing. Perhaps just using ‘moral behaviour” consistently is more precise? Or “a higher standard of morality”?

This has been revised as suggested (lines 160-162).

p. 7 line 152-154 - It strikes me that some of this explanation of past research is very abstract and difficult to understand or apply to this particular context without a concrete example. What would it look like for “third party observers” (still not very clearly defined who this would be…) to engage in a meaning making process around IRS and then conclude that IRS benefitted Indigenous people and Indigenous people are now held to a higher moral standard. Are there contemporary or historic examples to illustrate this? Perhaps adding one in earlier on (in the context of the very nice and clear description of the history of the IRS system?) would improve clarity throughout this intro? Once again Senator Lynn Beyak’s recent story comes to mind as a possibility to highlight?

As noted in previous comments to Reviewer 2, we have addressed the definitional issues earlier in the introduction, foreshadowed the relevance of meaning making, and included the example of Lynn Beyak (with some reluctance to give her further airtime that she doesn’t deserve!). We also clarified the meaning of the statement regarding meaning making (lines 170-171). We hope that as a result this paragraph is now clearer.

p. 9 line 185-187 - this is the most clearly I have understood what has been meant by “meaning making” previously. Perhaps you can more clearly explain you are simply asking people what was “learned” or what the “implications” were related to IRS for different groups? That may make your research questions more clear earlier on. Perhaps you could frame the intro more as something like “Learning about the IRS system for the first time has set off a process of reflection and wider meaning-making for non-Indigenous Canadians which may differ depending on several factors including if they are thinking about the implications/lesons for non-Indigenous versus Indigenous Canadians.” Or something like that. If this is a clear summary of the point of these studies it was not as clear as it should be earlier on.

We really liked Reviewer 2’s reframing and idea of putting a version of this suggested statement at the outset, and have done so (lines 55-60).

Page 10 lines 205-206 “meaning making condition did not significantly affect levels of racism reported” could be rephrased more clearly.

We have revised this statement to say “Levels of racism reported did not significantly vary across meaning making conditions” (lines 223-224).

P. 11 a figure depicting the mediation model would have been useful

We agree and have added this (Figure 1).

P. 12 lines 260-262 - this line suggests there actually WERE benefits bestowed via IRS…suggest change to “…through the benefits some participants may have believed were bestowed by the IRS system” to make this more clear, and less problematic.

We have reworded to clarify that these were participants’ perceptions… “by motivating some participants to justify the IRSs through their beliefs that benefits were bestowed by the IRS system” (lines 285-287).

p. 12. Line 262 “the Indigenous peoples” should just be “Indigenous peoples” or perhaps “Indigenous peoples in Canada” to be consistent and precise

This was a typo – thank you.

p. 13 line 265 I note you speak of just “observers” rather than “third party observers” here - perhaps the “third party” is unnecessary and could be removed elsewhere too?

As noted in an earlier response to Reviewer 2, we have clarified the terminology, and have avoided the use of the term third-party throughout except where it was directly relevant in terms of our expectations of non-Indigenous Canadians (as historical perpetrators) relative to the existing research in this area, and relative to Americans.

P. 13 - the rationale for Study 2 is beautifully described! Such an important issue that has not received enough attention in the discourse, scholarly or otherwise. I don’t mean to toot my own horn here but it bears mentioning that some of my own work is at least somewhat relevant to this study in particular. It is a qualitative study reporting Indigenous peoples’ own preferences for different representations of colonial history and we unpack some of the implications of more strength vs deficit based framings for Indigenous-settler intergroup relations in our discussion section: https://jspp.psychopen.eu/index.php/jspp/article/view/5185/5185.pdf We also (I believe…) encouraged more experimental work on this topic - which the authors are somewhat responding to with this study. To be clear, I am Not suggesting the authors need to cite this - but it may be a useful source to consult as the authors continue their work on this topic. It at least suggests that it is not only the media contributing to this discourse but Indigenous people themselves as well.

We agree that this paper was highly relevant to the point being made in Study 2, and have now included it ([24]). It is particularly relevant in the discussion section, and so has been highlighted in lines 566-575; 587-591.

P. 19 Wonderful to see this content analysis of the differential American/Canadian responses to the manipulation primes. This is an effective and compelling approach to treating qualitative data of this nature and considerably adds to our understanding of the overall story here. The X2 analysis and summary table bring this all together very effectively. They highlight for us the importance of recognizing that certain categories the authors had operationalized in the quantitative studies (e.g. “benefit finding”) can be understood in different and more nuanced ways, e.g. a more paternalistic “they should have learned to not make others suffer” vs. a more political / resistant “they learned not to trust outsiders”. Some discussion of how these findings may nuance prior qualitative findings and operationalizations of these concepts in the experimental components of the study would have been a welcome addition.

We are relieved to see the positive response to the inclusion of qualitative data. We have incorporated greater discussion of these findings to highlight the view of non-Indigenous Canadians that the IRS experience resulted in Indigenous peoples being more mistrusting of the motives of others, and what the potential implications are (lines 488-497).

Regarding coding, I did wonder if the authors could at least provide a rough picture of how long these responses were on average (character count average and range could work) and if they could at least tell the reader how many people coded these and if there was a process for resolving disagreement over two coders who differed on a specific code.

More information has been provided regarding the content coding of the data (lines 430-439). This includes information regarding the length of written responses, which did not differ across conditions.

p. 24 line 515 - I’m not sure saying that the different framings (strong vs vulnerable) is best described as having been “lost on perpetrator group members” - this suggests a more general indifference than the data warrant and could be phrased more specifically: these frames did not seem to affect what we measured. This does not rule out other implications those framings could have that are beyond the scope of what this study examined.

The reviewer is correct that this was not our intended meaning. We have reworded this statement to indicate that “framing the status of the victim group as strong versus vulnerable did not appear to alter non-Indigenous Canadians’ propensity to bestow benefits and apply moral expectations of Indigenous peoples” (lines 566-569).

p. 25 As mentioned above - this may be a worthwhile moment to consider acknowledging that Indigenous communities themselves engage in internal debate about how to represent themselves and the impact of colonial harms and the various pros and cons of different representations (e.g. victims vs resilient) for intergroup relations. Our qualitative work examining this in the context of an urban Indigenous community may be worth referencing here: https://jspp.psychopen.eu/index.php/jspp/article/view/5185/5185.pdf

Once again, this is obviously not a requirement and I leave it to the authors’ discretion.

As noted in an earlier response to Reviewer 2, we expanded this paragraph to more fully address these issues (lines 569-575).

P. 25 line 525 - I am slightly unclear how highlighting Indigenous strengths may lead to outgroups blaming them for having created the adverse conditions Indigenous people continue to face…perhaps this could be clarified? 

To provide clarity, we have included a Canadian example of this happening, and in particular in relation to reactions to the missing and murdered Indigenous women and girls inquiry that elicited victim blame when women demanded justice (lines 581-584).

However, on a related note and In part based on our work on this topic, I would encourage the authors to consider that there are other potential costs and benefits to emphasizing the success/resilience of Indigenous communities beyond the suggestion by outgroup members that they are “to blame” for what has happened to them. The impacts of this representation also may vary depending on their audience (Indigenous vs. non-Indigenous). For example resilience and resistance stories shared amongst Indigenous community members are a valuable source of in-group pride, but on the other hand may weaken claims for ongoing support from perpetrator outgroups: if you’re so resilient why do you still need our help? Likewise for representations of the “damage” inflicted in Indigenous communities via colonial policy. These representations can be drawn on as an excuse within Indigenous communities for certain negative behaviours, but alternatively these representations can be identity-protective in that they may help some Indigenous people better explain to themselves and other community members/family “why things are the way they are” in ways that attribute harms to external (what they did to us) rather than internal (who we are) factors. These possibilities may have implications for other parts of the authors’ findings. They at least suggest that there are more factors to consider than what outgroups will think about Indigenous peoples if they learn that they are resilient and strong.

We agree. Such possibilities are included in the rationale for Study 2 (lines 296-302), and have expanded on this in the context of the challenges faced by Indigenous advocates as they try to promote change (lines 576-584).

p. 25 line 531-533 - the line “The finding that modern racist beliefs were associated with perceiving Indigenous peoples to have benefited or learned from their past victimization and, as a result, they should be better as a people…” could perhaps be more nuanced to take into account what the authors found in the content analysis: sometimes participants felt that what Indigenous peoples may have “learned” from their experiences of victimization were self-protective norms of mistrust of outsiders etc. - this is not quite consistent with the more simplified notion I fear the authors occasionally imply (and that is implied here) that “learning from the experience” is the same as “benefitting from the experience”. Is someone who says “Indigenous people learned to become more vigilant and mistrustful of settlers as a result of the IRS system” saying that they “benefitted” from these schools in the same way senator Lynn Beyak has said Indigenous people “benefitted” from the IRS system? (https://www.cbc.ca/news/politics/senator-lynn-beyak-suffered-residential-schools-1.4042627) I would think not - the authors could perhaps do more to clarify the differences here in how participants were interpreting and responding to their “benefit finding” measure, as the content analysis of written responses suggested.

We agree that learning from the experience and benefit finding are distinct. However, the measure of benefit finding was in fact about deriving benefits such as being stronger, kinder, appreciating life more, motivated to succeed, and being a better person. The writing task was not intended to be a benefit finding manipulation, but rather a strategy for reflecting on the meaning of the experience for Indigenous peoples. For this reason, the qualitative analysis was able to highlight that participants not only identified ‘negative’ learnings, but also used the opportunity to distances themselves from the events (ie., by viewing them as a thing of the past). We hope the discussion for Study 2 is now clearer in this regard, given our expansion of non-Indigenous Canadians’ responses (lines 488-497). 

P. 25 - does this study have any limitations? None are noted. Issues of generalization at least are important to note, particularly if the authors wish to generalize from their findings to make rather grave pronouncements of the likely failure of reconciliation or healing in Canada…see below

We have expanded on the study limitations which were integrated throughout the general discussion in terms of alternative explanations and the absence of measures (lines 531-535; 578-580), the correlational nature of some of the data/models (lines 544). We have now added further comment on the sample and design limitations (lines 608-620).

p. 25 lines 539-541: The line “…such a relationship is unlikely to be achieved if addressing past wrongs is more likely to elicit psychological processes that serve the perpetrator group’s need to believe that justice has already been served.” Is a rather grim way to end the paper…are there no implications or findings implied by the paper that suggest a remedy? I encourage the authors to at least find a glimmer of hope to offer as their parting statement!

We agree that our ending is rather grim. And indeed, Canada and non-Indigenous Canadians do appear to miss many opportunities to bring about social change and equity in relation to Indigenous peoples. However, it isn’t hopeless – the increasing strength in the voices of Indigenous advocates and allies is noted (e.g., lines 573-575). And we’ve now on a different note that points to a greater need for positive intergroup contact, and the role of allies that do not espouse modern racist views (lines 617-624).

---

## [Decision Letter · Decision Letter 1]

10 May 2021

Non-Indigenous Canadians’ and Americans’ moral expectations of Indigenous peoples in light of the negative impacts of the Indian Residential Schools

PONE-D-20-31541R1

Dear Dr. Matheson,

We’re pleased to inform you that your manuscript has been judged scientifically suitable for publication and will be formally accepted for publication once it meets all outstanding technical requirements.

Kind regards,

Amy Michelle DeBaets, PhD

Academic Editor

PLOS ONE

Additional Editor Comments (optional):

Reviewers' comments:

Reviewer's Responses to Questions

**Comments to the Author**

1. If the authors have adequately addressed your comments raised in a previous round of review and you feel that this manuscript is now acceptable for publication, you may indicate that here to bypass the “Comments to the Author” section, enter your conflict of interest statement in the “Confidential to Editor” section, and submit your "Accept" recommendation.

Reviewer #1: All comments have been addressed

Reviewer #2: (No Response)

2. Is the manuscript technically sound, and do the data support the conclusions?

Reviewer #1: Yes

Reviewer #2: Yes

3. Has the statistical analysis been performed appropriately and rigorously? 

Reviewer #1: I Don't Know

Reviewer #2: Yes

4. Have the authors made all data underlying the findings in their manuscript fully available?

Reviewer #1: Yes

Reviewer #2: Yes

5. Is the manuscript presented in an intelligible fashion and written in standard English?

Reviewer #1: Yes

Reviewer #2: Yes

6. Review Comments to the Author

Reviewer #1: All suggestions have been adequately addressed. It would be helpful to have accessed the accurate number of children in the missing children registry for residential schools (currently 4118). As much accuracy as possibly helps stave off denial.

Reviewer #2: Dear Editor and Authors,

I have now had a chance to review the authors' responses to my initial review and have also reviewed their revised manuscript.

In general I’m very impressed with the revisions and feel everything is now much more clear. I am happy to recommend acceptance of this manuscript for publication, though draw the authors' attention to three minor wording issues I noticed on second review at the end of this brief response, after a few commendations.

The explanation of just world beliefs and connection to motivations to perceive benefits for victimized groups in the intro is much better, much more clear now!

Line 122 nice clear explanation of third party observers now.

Glad to see the (appropriately brief and negative!) reference to Lynn Beyak as well! I think that helps the illustration and to draw together a much more clear explanation now with a concrete example relevant to the context. Nice work.

The brief description of the coding process was excellent and increases readers’ confidence in these qualitative results.

The addition of the MMIWG example was very well-chosen and perfectly clarified what I thought was unclear, while also highlighting an important and relevant issue. Thank you!

I appreciate the authors’ response to my question about differences between “learning from the experience” and “benefitting from the experience” and am satisfied with their reasoning (and the clarifications in the text helped too).

I was also glad to see the authors felt our research on preferences for different representations of colonial history was relevant and useful for their analysis here. The additions that highlighted Indigenous movements towards shaping their own representations, resistance and resilience were very welcome as well and help the overall manuscript feel more balanced.

I really loved the new conclusion and felt it was so important to not only highlight the limitations of the study in a slightly expanded and more general way here (ironically, they make the study more hopeful!) but also point towards the wider context of social change in this complex context.

Three very small things I noticed on my second read that I wished to draw the authors’ attention to:

Line 50: I would suggest hedging this claim more to recognize the limits of generalization in the research being cited: “victim group group members are SOMETIMES or OFTEN perceived to have psychologically benefitted…” This is a possibility you are going to explore in a new context, not a foregone conclusion. You do this in the next line (“often expected”) and should do it here too.

Lines 60-61 - advise not using the phrase “assess this” twice in the same sentence

Line 426 I’m not sure this subtitle is properly worded? “Content analysis of written meaning derived” seemed a bit unclear?

Similarly in the title for Table 3 “written meaning made” is awkwardly phrased…

All in all, very well done and I look forward to seeing this work published!

7. PLOS authors have the option to publish the peer review history of their article (what does this mean?). If published, this will include your full peer review and any attached files.

Reviewer #1: No

Reviewer #2: **Yes: **Scott Neufeld

---

## [Editor Report · Acceptance letter]

12 May 2021

PONE-D-20-31541R1 

Non-Indigenous Canadians’ and Americans’ moral expectations of Indigenous peoples in light of the negative impacts of the Indian Residential Schools 

Dear Dr. Matheson:

I'm pleased to inform you that your manuscript has been deemed suitable for publication in PLOS ONE. Congratulations! Your manuscript is now with our production department. 

Kind regards, 

on behalf of

Dr. Amy Michelle DeBaets 

Academic Editor

PLOS ONE